# Evaluation of transplacental transfer of mRNA vaccine products and functional antibodies during pregnancy and infancy

Mary Prahl [1,2,11] ✉, Yarden Golan[3], Arianna G. Cassidy[4], Yusuke Matsui[5], Lin Li[4,6], Bonny Alvarenga [7], Hao Chen[6], Unurzul Jigmeddagva[6], Christine Y. Lin[4], Veronica J. Gonzalez[4], Megan A. Chidboy[4,6], Lakshmi Warrier[8], Sirirak Buarpung[4,6], Amy P. Murtha[6], Valerie J. Flaherman[1], Warner C. Greene[5,8,9], Alan H. B. Wu[10], Kara L. Lynch [10], Jayant Rajan[8] & Stephanie L. Gaw [4,6,11] ✉

Studies are needed to evaluate the safety and effectiveness of mRNA SARS-CoV-2 vaccination during pregnancy, and the levels of protection provided to their newborns through placental transfer of antibodies. Here, we evaluate the transplacental transfer of mRNA vaccine products and functional anti-SARS-CoV-2 antibodies during pregnancy and early infancy in a cohort of 20 individuals vaccinated during late pregnancy. We find no evidence of mRNA vaccine products in maternal blood, placenta tissue, or cord blood at delivery. However, we find time-dependent efficient transfer of IgG and neutralizing antibodies to the neonate that persists during early infancy. Additionally, using phage immunoprecipitation sequencing, we find a vaccine-specific signature of SARS-CoV-2 Spike protein epitope binding that is transplacentally transferred during pregnancy. Timing of vaccination during pregnancy is critical to ensure transplacental transfer of protective antibodies during early infancy.

Growing evidence has shown that pregnant individuals are at higher risk for SARS-CoV-2-related morbidity and mortality[1-4]. Despite this, vaccination uptake by pregnant individuals has been slower than the general population[5], in part because of maternal concern of adverse effects on the embryo or fetus, even with strong consensus recommendations for COVID-19 vaccination prior to or during pregnancy from several medical societies[6]. Pregnant individuals were excluded from initial vaccine trials, and complete data on safety, efficacy, optimal timing of the vaccine in pregnancy, or its impact on the fetus has been delayed[7], which may impact individual medical decision making. Current COVID-

19 vaccines in the United States include the mRNA vaccines BNT-162b2 and mRNA-1273, which target the SARS-CoV-2 Spike protein and stimulate protective immune responses[8,9]. In addition to protecting the mother against severe disease, vaccination during pregnancy may protect the newborn through passive transfer of maternal immunoglobulin. SARS-CoV-2 infection and vaccination during pregnancy produce an IgG response that is transferred to the fetus[10-16].

Prior studies have shown evidence of SARS-CoV-2 antigen exposure after maternal infection as demonstrated by both direct detection of the virus in delivery specimens, as well as production of

[1]Department of Pediatrics, University of California, San Francisco, CA, USA. [2]Division of Pediatric Infectious Diseases and Global Health, University of California, San Francisco, CA, USA. [3]Department of Bioengineering and Therapeutic Sciences, University of California, San Francisco, CA, USA. [4]Division of Maternal Fetal Medicine, Department of Obstetrics, Gynecology, and Reproductive Sciences, University of California San Francisco, San Francisco, CA, USA. [5]Gladstone Center for HIV Cure Research, Gladstone Institute, San Francisco, CA, USA. [6]Center for Reproductive Sciences, Department of Obstetrics, Gynecology, and Reproductive Sciences, University of California San Francisco, San Francisco, CA, USA. [7]Department of Neurology, University of California, San Francisco, CA, USA. [8]Department of Medicine, University of California, San Francisco, CA, USA. [9]Departments of Microbiology and Immunology, University of California, San Francisco, CA, USA. [10]Department of Laboratory Medicine, University of California, San Francisco, CA, USA. [11]These authors jointly supervised this work: Mary Prahl, Stephanie L. Gaw. ✉e-mail: mary.prahl@ucsf.edu; Stephanie.Gaw@ucsf.edu

fetal/infant anti-SARS-CoV-2 IgM—as IgM is not transplacentally transferred and is indicative of a fetal/infant endogenous immune response to antigen[16–20]. However, further studies are needed to determine if mRNA vaccination during pregnancy leads to fetal SARS-CoV-2 antigen exposure.

In addition to the overwhelming evidence of maternal protection against COVID-19 after SARS-CoV-2 vaccination in pregnancy[21–25], evidence of newborn protection might help address maternal concerns about adverse effects. However, detailed studies of the transplacental transfer of vaccine products and vaccine-related antibody dynamics, functional properties, and persistence during infancy of transferred SARS-CoV-2 antibodies are needed to provide such evidence.

We examined the transplacental transfer of mRNA vaccine products and humoral responses using samples from pregnant individuals and their infants vaccinated with either BNT-162b2 or mRNA-1273 mRNA vaccine during pregnancy.

## Results

### Cohort

We evaluated 20 pregnant individuals who received COVID-19 mRNA vaccines during pregnancy and their infants. Participants were vaccinated between December 2020 and April 2021. Gestational age at first dose ranged from 13 weeks to 40 weeks (mean 31.2, SD 5.9 weeks). Nineteen participants delivered live, singleton infants between January 2021 through April 2021 at gestational ages ranging from 37.4 to 41.1 weeks (mean 39.2, SD 1.1 weeks). One participant who was vaccinated at 13 weeks had a termination of pregnancy due to a lethal skeletal dysplasia of genetic etiology at 20.4 weeks. Eight participants received BNT-162b2 (Pfizer-BioNTech) and twelve received mRNA-1273 (Moderna) vaccines. Eighteen participants received both vaccine doses prior to delivery, and two participants received the second dose after delivery. The time from the first mRNA vaccine dose ranged from 6 to 97 (mean 51, SD 24.3) days prior to delivery, time from the second dose ranged from 2 to 75 (mean 32, SD 21.3) days prior to delivery, and in two participants 15 and 21 days after delivery (Table S1). No participants received a 3rd dose of vaccine prior to delivery. Infants born to vaccinated mothers were followed up at convenience timepoints ranging from age 3 weeks to 15 weeks of life (mean 8.3, SD 3.2). All participants were negative for prior COVID-19 infection by self-reported

survey and by maternal anti-nucleocapsid IgG screening at delivery. Further demographic data is detailed in Table S1.

### Vaccine mRNA products do not cross the placenta at readily detectable levels

To determine the transplacental transfer of mRNA vaccine-derived products, we examined available maternal blood at delivery, placenta tissue, and cord blood for Spike protein by western blot and vaccine mRNA by PCR. All available delivery samples (maternal blood, placental tissue, and cord blood) were negative for Spike protein by western blot (Supp Fig. 1, Supp Table 3) and did not have detectable levels of vaccine mRNA by PCR (Suppl Table 3). Together, this indicates that products of mRNA vaccination do not reach the fetus after vaccination during pregnancy at readily detectable levels.

### mRNA vaccination in pregnancy leads to a robust antibody response

Similar to prior studies[14,15,26], we found that mRNA vaccination during pregnancy led to an increase in anti-SARS-Cov-2 IgG following dose 1 ($n = 7$, mean 388.6, SD 224.8 RFU) and an even further robust increase after vaccination dose 2 ($n = 12$, mean 3214, SD 1383 RFU). Anti-SARS-CoV-2 IgM ($n = 7$, mean 53.3, SD 50.2 RFU) was detected in two maternal participants following dose 1, but only 1 participant following dose 2 ($n = 12$, mean 23.8, SD 17 RFU, Fig. 1).

### Vaccine-induced anti-SARS-CoV-2 IgG and neutralizing antibodies are transplacentally transferred

We then evaluated the transplacental transfer of maternally-derived anti-SARS-CoV-2 IgG antibodies to their infants. Maternal blood at delivery was available in 19/20 participants and cord blood was available in 17/20 participants. Anti-SARS-CoV-2 IgG was detectable in 94.7% (18/19) of maternal blood samples at delivery (mean 3235, range [10, 7811] RFU). Anti-SARS-CoV-2 IgG was detectable in 88.2% (15/17) cord blood samples (mean 2243, range [2, 4959] RFU). One participant received one mRNA vaccine dose 9 days prior to delivery, and both the maternal and cord blood were negative for IgG at the time of delivery. Another participant received two doses of mRNA vaccine (23 and 2 days) prior to delivery and maternal blood was positive at 55 RFU (positive cutoff >50 RFU), however, cord blood IgG was negative (Fig. 2A). Maternal and

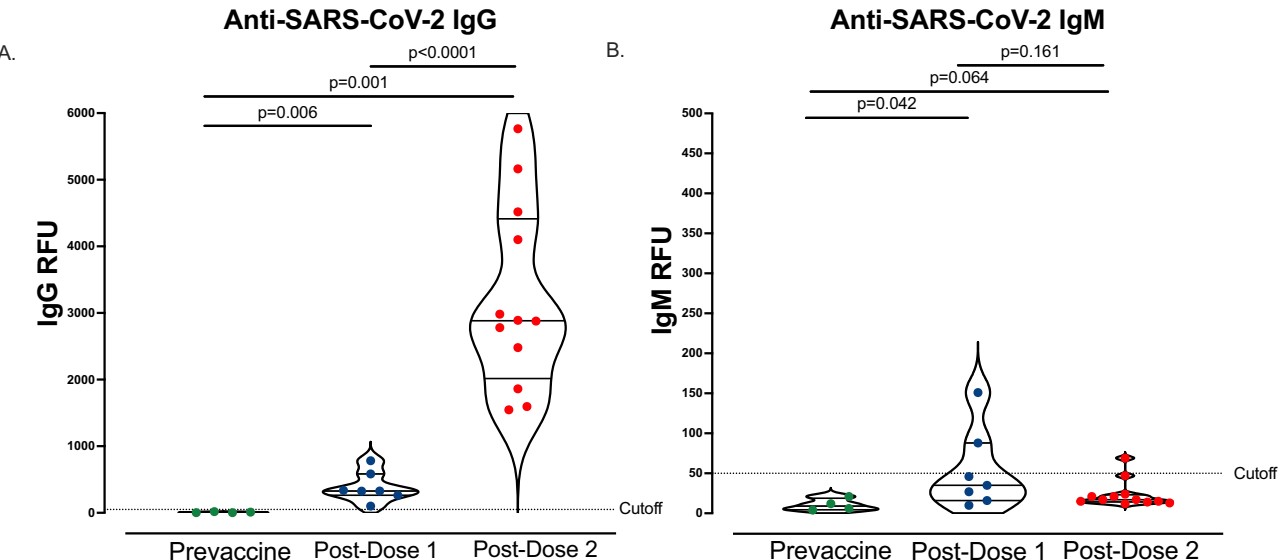

**Fig. 1 | Anti-SARS-CoV-2 IgG and IgM antibody responses following vaccination.**
**A** Maternal plasma anti-SARS-CoV-2 RBD/N IgG antibody relative fluorescence units (RFU) levels prior to vaccination ($n = 4$), 3–4 weeks post-dose 1 ($n = 7$), and 4–8 weeks post-dose 2 ($n = 12$). **B** Maternal plasma anti-SARS-CoV-2 RBD/N IgM (RFU) levels prior to vaccination ($n = 4$), 3–4 weeks post-dose 1 ($n = 7$), and 4–8 weeks post-dose 2 ($n = 12$). Wilcoxon rank-sum testing. Data represent median ± quartiles, two-sided $p$ values were calculated for all test statistics. Source data are provided as a source data file.

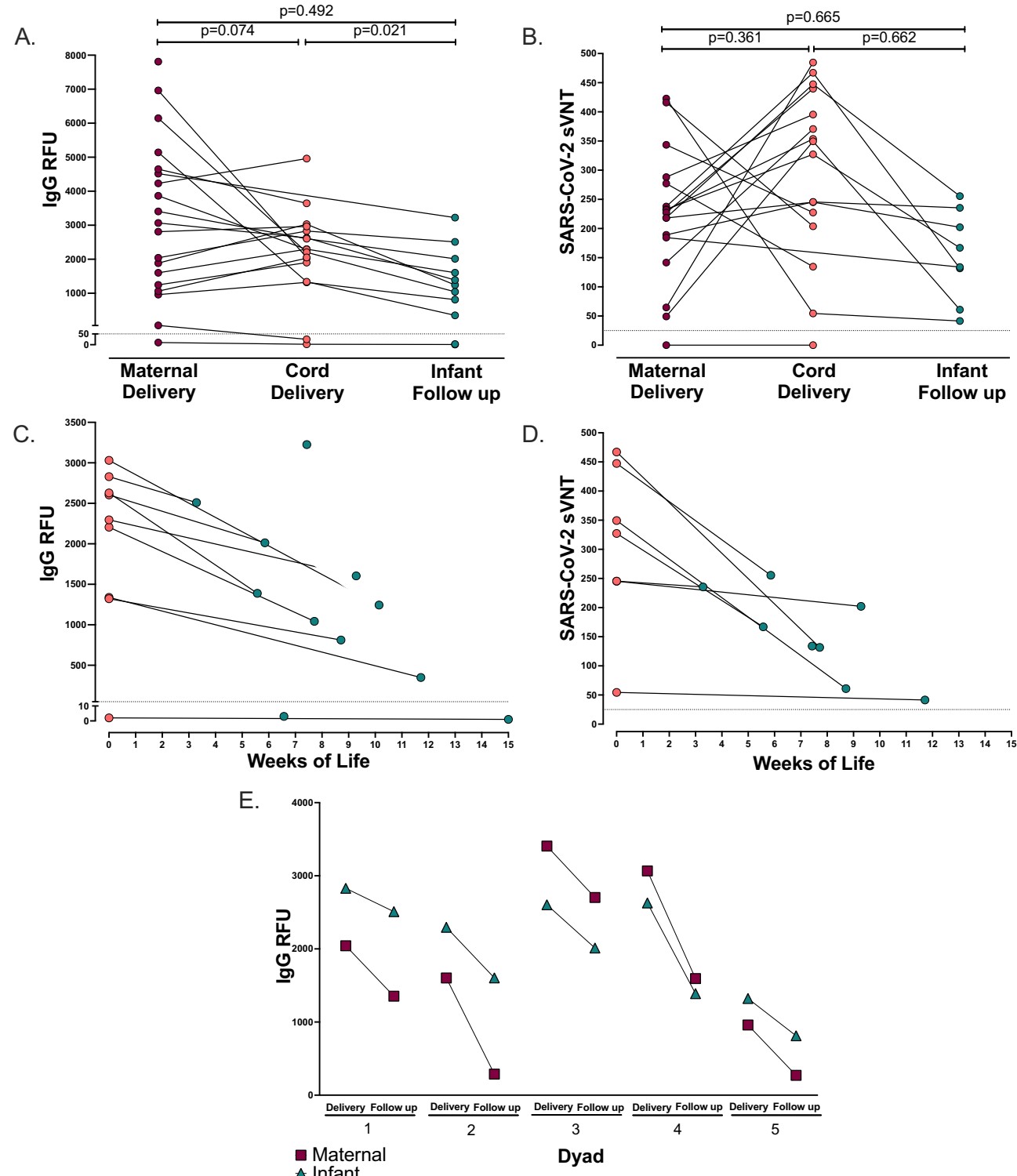

**Fig. 2 | Paired maternal, cord, and infant IgG and neutralization antibodies.**
**A** Paired maternal plasma at delivery (*n* = 19), cord plasma (*n* = 17), and infant follow-up (*n* = 11) by anti-SARS-CoV-2 RBD/N IgG antibody relative fluorescence units (RFU), (Spearman's rank correlation, dotted line indicates positive cutoff value of 50 RFU). **B** Paired maternal plasma at delivery (*n* = 17), cord plasma (*n* = 16), and infant follow-up (*n* = 8) by SARS-CoV-2 label-free surrogate neutralization assay (sVNT), (Spearman's rank correlation, dotted line indicates positive cutoff value of 25).

**C** Paired cord plasma (*n* = 9) and infant follow-up plasma (*n* = 11) anti-SARS-CoV-2 RBD/N IgG by weeks of life. **D** Paired cord plasma (*n* = 7) and infant follow-up plasma (*n* = 8) label-free surrogate neutralization assay (sVNT) by weeks of life. **E** Paired maternal plasma at delivery (*n* = 5), cord plasma (*n* = 5), and paired maternal follow-up (*n* = 5) and infant follow-up plasma (*n* = 5) anti-SARS-CoV-2 RBD/N IgG. Two-sided *p* values were calculated for all test statistics. Source data are provided as a source data file.

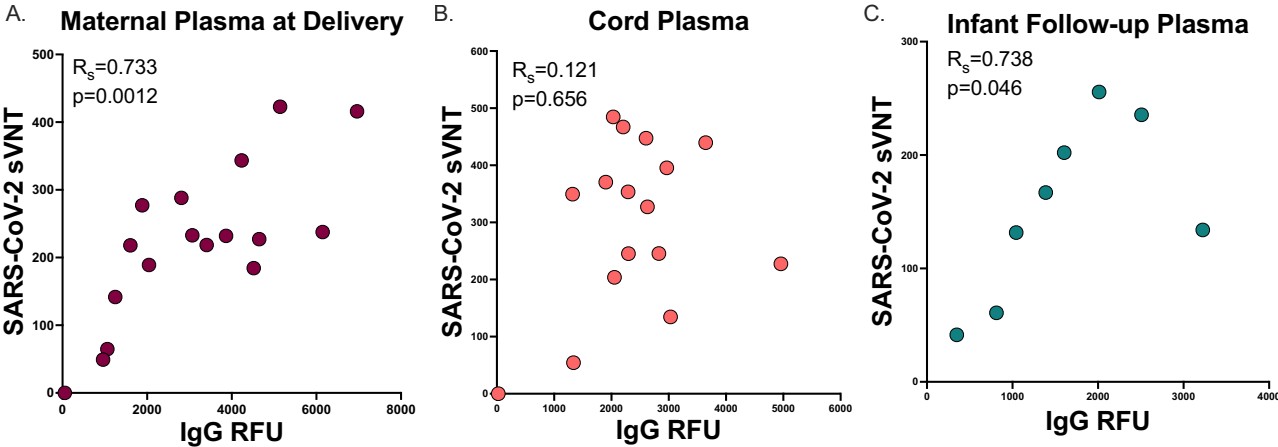

**Fig. 3 | Neutralization to IgG antibody correlation. A.** Maternal plasma at delivery ($n = 17$). **B** Cord plasma ($n = 16$). **C** Infant follow-up plasma ($n = 8$) SARS-CoV-2 label-free surrogate neutralization assay (sVNT) by anti-SARS-CoV-2 RBD/N IgG relative fluorescence units (RFU) correlation (Spearman's rank correlation). Two-sided $p$ values were calculated for all test statistics. Source data are provided as a source data file.

cord blood anti-SARS-CoV-2 IgG levels were moderately correlated, but not statistically significant ($p = 0.074$, $R_s = 0.446$, Fig. 2A). All cord blood samples were anti-SARS-CoV-2 IgM negative.

We next evaluated the transplacental transfer of neutralizing antibody titers by a label-free surrogate neutralization assay (sVNT) from mother to cord blood. Maternal and cord blood at delivery had robust neutralizing responses (maternal $n = 17$, mean 220.2, range [0, 422]. Cord blood $n = 16$, mean 296.6, range [0, 485], Fig. 2B). All mother-infant dyads with positive IgG serology at delivery had detectable transplacental transfer of neutralizing antibodies with the exception of one pair in which the mother was borderline IgG positive at delivery and cord blood was negative, for which both maternal and cord blood were negative for neutralizing titers (Fig. 2B). However, maternal and cord blood neutralizing titers were not significantly correlated ($p = 0.361$, $R_s = -0.244$, Fig. 2B). Taken together, this indicates that maternal mRNA vaccination induces functional neutralizing antibodies that are transferred to the infant.

**Maternally-derived vaccine-induced anti-SARS-CoV-2 IgG and neutralizing antibodies persist through early infancy**

A subset of infants was sampled at convenience timepoints during follow-up (infant $n = 11$, weeks of life range [3,15] mean 8.3 weeks). Anti-SARS-CoV-2 IgG levels were positive in 81.8% of infants at follow-up (9/11 infants, mean 1290, range [1, 3225] RFU, Fig. 2A), with one infant still IgG positive at 12 weeks of age (Fig. 2C). The two infants that were IgG negative at follow-up were both born to mothers who received only one vaccine dose prior to delivery (6 and 9 days prior to delivery, respectively). One of these infants did not have paired maternal or cord blood available at the time of delivery for comparison, and the other was IgG negative in cord blood. Maternal and infant follow-up anti-SARS-CoV-2 IgG levels were not significantly correlated; however, cord blood and infant follow-up IgG levels were significantly associated ($p = 0.492$, $R_s = 0.249$ and $p = 0.021$, $R_s = 0.76$, respectively, Fig. 2A). All infants were IgM negative at the time of follow-up.

All infants with available IgG positive samples at follow-up had detectable neutralizing titers ($n = 8$, mean 154, range [41–256], Fig. 2B). Maternal and infant follow-up neutralizing titers were not significantly correlated, as well as cord and infant follow-up neutralizing titers ($p = 0.665$, $R_s = -0.191$ and $p = 0.662$, $R_s = 0.214$, respectively, Fig. 2B).

To compare the rate of decay of IgG antibody levels in mothers and their infants, we evaluated 5 dyads with paired maternal and infant blood samples on the same day at the time of follow-up (range 3–9 weeks post-delivery). Maternal antibody IgG levels decreased faster in mothers than infants (mean delta −974 RFU and −670 RFU,

respectively. Figure 2E) at the follow-up timepoint. Taken together this indicates, maternally-derived functional vaccine-induced antibodies persist at high levels in newborns through early infancy during a critical time of immune vulnerability and may decay slower than maternal IgG antibodies.

**Vaccine-induced antibody timing and transplacental facilitated transfer**

We assessed the relationship of anti-SARS-CoV-2 IgG levels to neutralizing antibody levels. We found a strong correlation between IgG and neutralizing titers in maternal plasma at delivery ($R_s = 0.744$, $p = 0.0012$) and infant follow-up ($R_s = 0.738$, $p = 0.046$) timepoints, but no significant association between IgG and neutralizing titers in cord blood ($R_s = 0.121$, $p = 0.656$, Fig. 3).

We then evaluated the impact of timing of vaccination on maternal antibody levels at delivery. Consistent with the known kinetics of SARS-CoV-2 antibody responses[16,27], two participants received their first dose of vaccine <30 days prior to delivery had low or absent levels antibody levels at delivery and were excluded from analysis. We found maternal IgM levels at delivery were statistically significantly correlated with time since vaccine dose 1 ($R_s = -0.846$, $p < 0.0001$, Fig. 4A). In addition, maternal IgG levels at delivery were significantly correlated with time since dose 1 ($R_s = -0.866$, $p < 0.0001$) and gestational age at delivery ($R_s = 0.689$, $p = 0.002$, Fig. 4B, C). We then evaluated neutralizing titers in those participants with known detectable IgG levels at delivery and found that maternal neutralizing titers was significantly correlated with days since vaccine dose 1 ($R_s = -0.706$, $p = 0.002$). However, maternal neutralizing titers at delivery was not associated with gestational age at dose 1 ($R_s = 0.288$, $p = 0.279$, Fig. 4D, E).

To assess facilitated antibody transfer, we evaluated cord-to-maternal antibody IgG and neutralization titer ratios by time since vaccination and gestational age in all participants with available samples. We found that cord-to-maternal IgG and neutralization titer transfer ratios were not significantly correlated ($R_s = 0.257$, $p = 0.354$, Fig. 4F). However, IgG ratios were highly correlated with both time since first maternal vaccination dose and gestational age at first dose ($R_s = 0.917$, $p < 0.0001$ and $R_s = -0.739$, $p = 0.002$, respectively, Fig. 4G, H). In contrast, neutralization titer cord-to-maternal ratios by time since first vaccination dose and gestational age at first dose were not significantly associated ($R_s = 0.366$, $p = 0.179$ and $R_s = -0.032$, $p = 0.913$, respectively, Fig. 4I, J). Together, this may indicate that timing of vaccination in pregnancy is critical for maternal-fetal antibody transfer, and functional neutralizing antibodies are

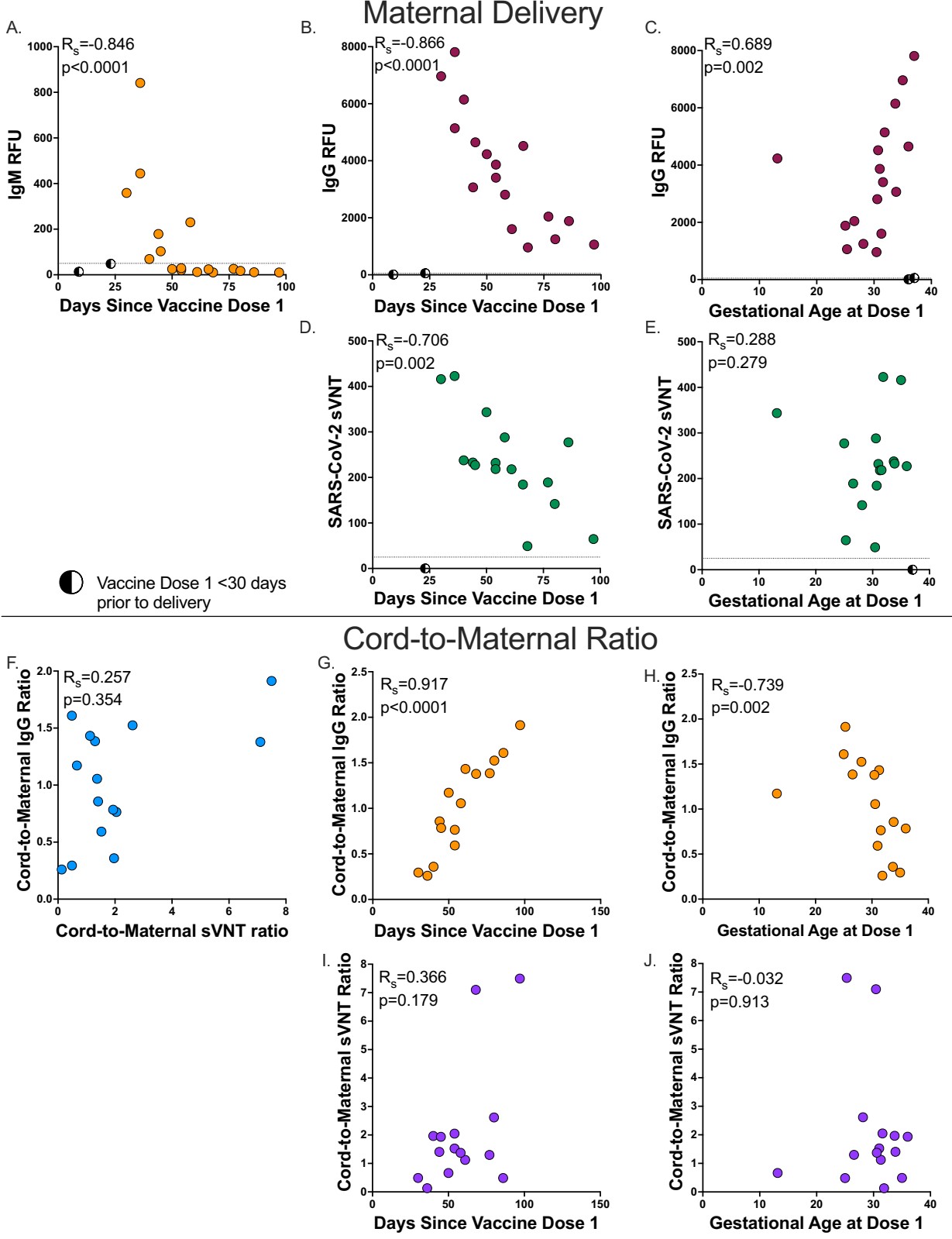

differentially transferred to the fetus as compared to total anti-SARS-CoV2 IgG during gestation.

**mRNA vaccination in pregnancy leads to a unique SARS-CoV2 Spike protein antibody epitope binding signature**

We next investigated antibody linear epitope binding and transplacental transfer using the PhIP-seq/VirScan SARS-CoV-2 Spike protein phage display array in mother-infant dyads at the time of birth (Fig. 5). We found that timing of vaccination was important for the transplacental transfer of Spike protein epitope binding antibodies. Two mother-infant dyads had minimal Spike protein-specific epitope binding. The first dyad only received one dose of mRNA vaccine 9 days prior to delivery, and the other dyad received the second vaccine dose 2 days prior to delivery.

**Fig. 4 | Maternal delivery and Cord-to-maternal antibody transfer ratios timing. A** Maternal delivery anti-SARS-CoV-2 RBD/N IgM antibody level by days since vaccine dose 1 ($n = 17$, dashed line indicates positive cutoff >50 relative fluorescence units (RFU)). **B** Maternal delivery anti-SARS-CoV-2 RBD/N IgG antibody level by days since vaccine dose 1 ($n = 17$, dashed line indicates positive cutoff >50 RFU). **C** Maternal delivery anti-SARS-CoV-2 RBD/N IgG antibody level by gestational age at vaccine dose 1 ($n = 17$, dashed line indicates positive cutoff >50 RFU). **D** Maternal delivery SARS-CoV-2 label-free surrogate neutralization assay (sVNT) antibody titer days since vaccine dose 1 ($n = 16$, dashed line indicates positive cutoff >25). **E** Maternal delivery SARS-CoV-2 label-free surrogate neutralization assay (sVNT) antibody titer by gestational age at vaccine dose 1 ($n = 16$, dashed line indicates positive cutoff >25). **A–E** Samples with vaccine dose 1 < 30 days prior to delivery

(indicated by half circles) excluded from presented analysis. **F** Cord-to-maternal anti-SARS-CoV-2 RBD/N IgG antibody transfer ratio by cord-to-maternal label-free surrogate neutralization assay (sVNT) antibody transfer ratio ($n = 15$). **G** Cord-to-maternal anti-SARS-CoV-2 RBD/N IgG antibody transfer ratio by days since vaccine dose 1 ($n = 15$). **H** Cord-to-maternal anti-SARS-CoV-2 RBD/N IgG antibody transfer ratio by gestational age at vaccine dose 1 ($n = 15$). **I** Cord-to-maternal SARS-CoV-2 label-free surrogate neutralization assay (sVNT) antibody transfer ratio by days since vaccine dose 1 ($n = 15$). **J** Cord-to-maternal SARS-CoV-2 label-free surrogate neutralization assay (sVNT) antibody transfer ratio by gestational age at vaccine dose 1 ($n = 15$). Spearman's rank correlation. Two-sided $p$ values were calculated for all test statistics. Source data are provided as a source data file.

We found high levels of SARS-CoV-2 Spike protein epitope binding in 4 major peaks we designate as regions 1–4 (Fig. 5A). Region 1 overlays the carboxy terminal of the N-terminal domain. Region 2 overlaps with key residues for the S1/S2 cleavage site. Regions 3 and 4 are within S2, flanking the fusion loop and the transmembrane portion of the Spike protein, respectively. However, we found minimal binding in the receptor binding domain (RBD) of Spike protein. Prior evaluation using the PhIP-seq/VirScan SARS-CoV-2 epitope phage array during SARS-CoV-2 infection demonstrated similar binding in regions 3 and 4, however, in SARS-CoV-2 infection there was minimal binding in regions 1 and 2 demonstrating that antibody epitope binding in these regions may be unique to vaccination during pregnancy[28]. In addition, there is proportional representation of linear epitope binding across the SARS-CoV-2 Spike protein proteome between mothers and infants (Fig. 5B). Taken together, SARS-CoV-2 antibody linear epitope binding after vaccination during pregnancy shows similar patterns, with multiple immunodominant regions found in the majority of mothers and infants. Some of these regions are unique to vaccination during pregnancy and not observed during natural infection in non-pregnant adults[28–30].

## Discussion

Among 20 women who received the COVID-19 mRNA vaccine during pregnancy, our study found no evidence of transplacental transfer of mRNA vaccine products at clinically relevant levels but did find high levels of functional vaccine-derived antibodies that transferred to the infant at delivery and persisted during early infancy. In addition, we identified high levels of epitope binding in two regions of Spike protein that may be unique to SARS-CoV-2 vaccination[28]. These data may address some of the many unanswered questions regarding COVID-19 vaccination in pregnancy, including (1) the dynamics of antibody production in the pregnant immune state and (2) the optimal timing of immunization in pregnancy to impart passive immunity to the newborn during the vulnerable first few weeks of infancy.

Uptake of COVID-19 vaccination in pregnancy has been slow[5], and reasons for vaccine hesitancy are likely multifactorial—but theoretical concerns that vaccine mRNA products could cross the placenta have been raised. We found no evidence of mRNA vaccine products in any of our delivery samples at readily detectable levels, consistent with a recent study that did not find evidence of mRNA vaccine-related Spike protein in placenta tissue by immunohistochemistry after maternal vaccination[31]. Other recent studies have also evaluated for the persistence of mRNA vaccine products after SARS-CoV-2 vaccination and have found the presence of mRNA vaccine products in serum up to 1 month after the first vaccine dose and a week after the second vaccine dose[32,33], or in local lymph nodes up to 60 days following the second vaccine dose[34].

In addition, we examined the fetal immune responses to SARS-CoV-2 vaccine antigens to determine any potential clinically relevant transplacental transfer of vaccine products that could have occurred either at levels below the level of sensitivity of our assays or out of the known timeframe of pharmacokinetic persistence of mRNA vaccine

products in humans after vaccination—as we evaluated samples at delivery ranging from days to months following vaccine doses. Consistent with other studies[17,35], no infants in our study had a fetal immune response to Spike protein as demonstrated by a negative anti-SARS-CoV-2 IgM antibody in cord blood and infant follow-up samples. This further supports the lack of transfer of vaccine products, as only IgG is transferred from the mother to the infant. IgM production would indicate an endogenous fetal immune response which has rarely been seen in natural infection with SARS-CoV-2 during pregnancy[16,36–38], including in our prior study of a premature infant who seroconverted to IgM positivity following a vertical transmission of SARS-CoV-2 with viral positivity in maternal blood, infant meconium, and infant nasopharyngeal swabs[16]. Together, this provides additional reassurance that mRNA vaccination is safe during pregnancy.

We found that the timing of immunization during pregnancy is important to ensure transplacental transfer of protective antibodies to the neonate, and during critical windows of immune vulnerability during early infancy. As seen in prior work showing robust immune responses to mRNA vaccination and transplacental IgG transfer during pregnancy[14,15,26,39–42], we found high levels of IgG after two doses of mRNA vaccine that transferred from the mother to the infant. In addition, as previously demonstrated with larger cohort studies[41–43], completion of the vaccination series well before delivery was important to ensure transfer of antibodies to the infant. Two mothers only received one vaccine dose prior to delivery and did not transfer antibodies as demonstrated by the lack of antibodies in cord (in one with available cord blood) and in both infants at follow-up. In addition, neutralizing antibodies were not transferred in a mother who received her second dose of vaccine 2 days prior to delivery. All evaluated mothers who received both doses during pregnancy and with the second dose >9 days prior to delivery transferred IgG and neutralizing antibodies to their infants. Consistent with early studies of antibody transfer after COVID-19 vaccination in pregnancy, most of our participants were vaccinated in the third trimester of pregnancy. Larger studies of individuals vaccinated prior to pregnancy, in the first and second trimester, and after boosting doses are needed to understand persistence and waning of vaccine-induced immune responses—for the protection of the mother during pregnancy, as well as transfer of protective antibodies to the neonate.

In addition, we believe we are the first to report that infants in the first few months of life continued to have maternal vaccine-derived anti-SARS-CoV-2 antibodies that were functional as demonstrated by high levels of neutralizing antibodies presenting infants up to 12 weeks of age. In support of transplacental antibody persistence, a recent study showed that half of infants at 6 months of age still had SARS-CoV-2 IgG after maternal vaccination[44]. This is similar to known persistence of maternally-derived antibodies from other vaccinations including pertussis, rubella, varicella[45–47]. In addition, we have previously found persistence of anti-SARS-CoV-2 IgG antibodies in infants after natural infection up to 6 months[16]. However, the functional capability of these antibodies as compared to anti-SARS-CoV-2 vaccination-derived antibodies is unknown. Limitations of our study included a small subset of

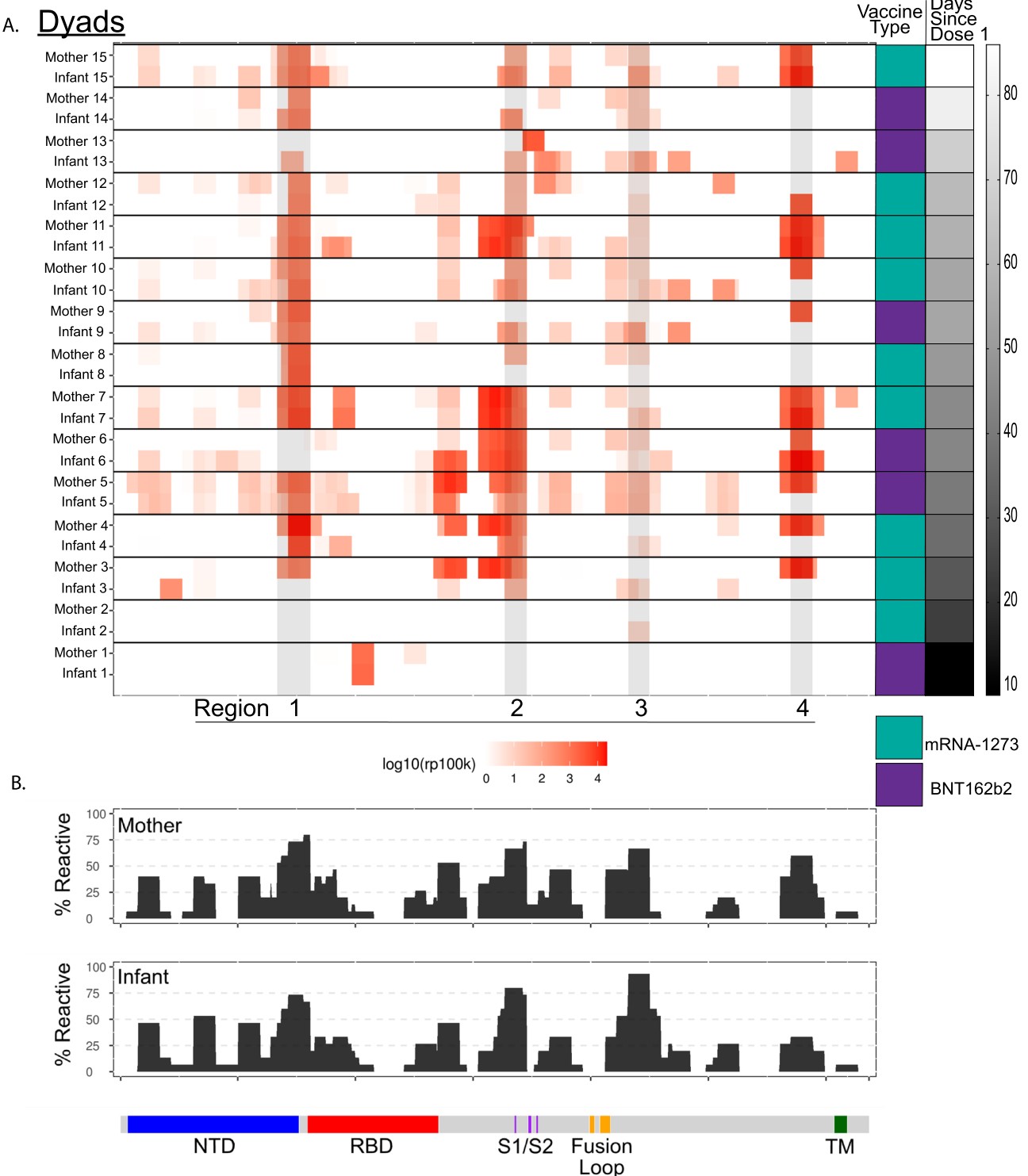

**Fig. 5 | PhIP-seq/VirScan paired maternal and cord SARS-CoV-2 Spike protein epitope binding. A** Heatmap displaying results of significant enriched (*p* < 0.001) linear SARS-CoV-2 Spike protein epitope binding from 15 paired mother-infant dyads in maternal plasma at delivery and cord plasma by vaccine type and time since vaccine dose 1. Areas of high cumulative epitope binding designated by regions 1–4. **B** Cumulative fold enrichment of mothers and infants linear SARS-CoV-2 Spike protein epitope binding. Counts per 100,000 reads for all peptides were modeled against the distribution of rp100k in healthy control samples modeled as normally distributed. One-sided calculated *p* values were corrected for multiple hypothesis using the Benjamini–Hochberg method. Any peptide with a corrected *p* value of <0.001 was considered significantly enriched over the healthy background. NTD = N-terminal domain, RBD = receptor binding domain, S1 = Spike 1 subunit, S2 = Spike 2 subunit, TM = Transmembrane. Source data are provided as a source data file link.

infants with samples available for analysis. Further evaluation of the longitudinal persistence of maternal vaccine-derived antibodies during infancy in larger cohorts will be critical to determine optimal timing of COVID-19 vaccination in infancy.

Evaluation of paired maternal and baby samples at post-partum follow-up timepoints showed a faster decline in maternal IgG antibody levels than infants, suggesting that maternally-derived antibody may be more durable in infants. Differences in renal excretion and neonatal

Fc receptor (FcRn) expression, which is involved in antibody degradation[48] in the infant as compared to adults, could underly these differences and should be explored further.

Consistent with observations in non-pregnant adults, we found that IgG levels in mothers at delivery, and at infant follow-up were highly correlated with neutralizing titers[49]. However, cord blood IgG levels did not correlate with neutralizing titers. Moreover, IgG cord-to-maternal ratios, which represent a proxy of maternal to fetal antibody transfer, were highly correlated with timing of vaccination (gestational age and days since the first dose), but cord-to-maternal neutralizing titer ratios were not significantly associated with cord-to-maternal IgG ratios, time since vaccination, nor gestational age. During gestation, there is facilitated transfer of maternally-derived antibodies through the binding of the neonatal Fc receptor in the syncytiotrophoblast layer[50]. Differences in glycosylation[51,52], FcR/FcRn binding affinity[26,53], preferential IgG subclass transfer[54,55] may be different in functional neutralizing antibodies as compared to total IgG antibody transfer. However, a limitation of this study is the majority of participants were vaccinated in the third trimester. Further investigations on factors that influence the transport of functional antibodies across trimesters are needed to understand antibody dynamics and optimal transfer of protective antibodies to infants.

Using a PhIP-seq/VirScan SARS-CoV-2 Spike protein phage array, we were able to compare linear epitope antibody binding in mothers and their infants. Consistent with IgG and neutralizing antibody evaluation, timing of vaccination was critical to ensure the transplacental transfer of antibodies to the infant. In addition, we identified unique regions of antibody epitope binding in our vaccinated pregnancy cohort that were not identified using the same phage library in a prior evaluation of a cohort of SARS-CoV-2 infected non-pregnant individuals[28]. One of these regions included the carboxy terminal of the N-terminal domain, with other work having shown that the N-terminal domain is targeted by neutralizing antibodies against Spike protein[56]. We did not see significant binding in the receptor binding domain (RBD), which may be attributable to the fact that the phage display library displayed short, linear peptides while antibodies targeting RBD are known to target conformational epitopes. Last, we found that the same immunodominant regions targeted by antibodies targeting the Spike protein in both mothers and infants. Though this may indicate a unique epitope binding signature after SARS-CoV-2 vaccination as compared to infection, key alterations in functional antibody responses during pregnancy could also underly these differences. During pregnancy, the immune system is poised towards regulatory responses in order to tolerate the semi-allogenic fetus. This may impact a number of functional immune responses to antigen exposure including following vaccination. Recent work has demonstrated differences in SARS-CoV-2 mRNA vaccine-induced antibody effector functions and FcR binding capacity in pregnant individuals compared to non-pregnant controls[26], as well as potential reduced mRNA vaccine-associated neutralizing activity to SARS-CoV-2 ancestral virus and variants of concern[57]. Further work is needed to understand differences in antibody formation and function after vaccination and infection in pregnancy.

COVID-19-related morbidity and mortality in pregnancy remain unacceptably high[58] despite vaccination roll out in late 2020. In summary, this work provides further evidence that mRNA vaccination is safe in pregnancy, and demonstrates that it generates time-dependent protective, functional antibody responses in mothers and infants that persist during early infancy.

## Methods
### Cohort and sample collection
The University of California San Francisco (UCSF) institutional review board approved the study (20-32077). Written informed consent was obtained from all participants. We enrolled 20 female pregnant individuals who were vaccinated with either BNT-162b2 or mRNA-1273 mRNA vaccines (age ranges Table S1). Pregnant individuals were followed through delivery, and their infants were followed up to 12 weeks of life. No participants reported a history of known COVID-19 infection by self-reported survey. Maternal blood was collected during pregnancy (pre-vaccine, 3–4 weeks post-dose 1, 4–8 weeks post-dose 2). During delivery, maternal blood, placenta tissue, and cord blood was collected. Infant follow-up blood was collected at convenience timepoints. Maternal, cord, and infant follow-up blood was collected in EDTA collection tubes, and the sampling processing protocols were consistent for all blood samples. Whole blood was immediately added to RNAlater in a 1:1.3 ratio. Plasma was isolated from whole blood by centrifugation and immediately cryopreserved. Full-thickness placental biopsy was collected within 1 h of delivery, washed three times with phosphate-buffered saline, and preserved in RNAlater.

### SARS-CoV-2 plasma serology
We used a clinically validated serologic assay to assess anti-SARS-CoV-2 IgM and IgG[16,27,59,60]. This assay has previously used in a pregnancy cohort of SARS-CoV-2 infected individuals to detect a vertical transmission of SARS-CoV-2 after maternal infection[16]. Anti-SARS-CoV-2 plasma IgM and IgG antibodies were measured for combined anti-spike protein receptor binding domain (RBD) and anti-nucleocapsid protein (NP) antigens using the Pylon 3D automated immunoassay system (ET Healthcare, Palo Alto, CA). In brief, quartz glass probes are pre-coated with either affinity purified goat anti-Human IgM (IgM capture) or Protein G (IgG capture) are dipped into diluted patient sample. Samples are washed, and then the probe is dipped into the assay reagent containing both biotinylated recombinant spike protein receptor binding domain (RBD) and nucleocapsid protein (NP). After washing, the probe is incubated with a Cy®5-streptavidin (Cy5-SA) polysaccharide conjugate reagent, allowing for cyclic amplification of the fluorescence signal. The background corrected signal is reported as relative fluorescent units (RFU) which is proportional to the amount of specific antibodies in the sample allowing for quantification. Levels of IgM and IgG were considered positive if >50 relative fluorescence units. In addition, we screened maternal delivery samples for the presence of anti-nucleocapsid IgG to determine any potential prior SARS-CoV-2 infection by human anti-nucleocapsid IgG ELISA kit (RayBiotech, Peachtree Corners, Georgia) according to manufacturer's instructions. Samples were run in technical replicates, and less than 9 units/mL was considered negative.

### SARS-CoV-2 neutralizing assay
SARS-CoV-2 antibody neutralization titers were measured using a label-free surrogate neutralization assay (LF-sVNT) previously described[49]. Briefly, the method measures the binding ability of recombinant RBD (Sino Biological, Wayne, PA) coated onto sensing probes (Gator Bio, Palo Alto, CA) to recombinant ACE2 (Sino Biological, Wayne, PA) after neutralizing RBD with SARS-CoV-2 antibodies in serum. Measurements were done using a thin-film interferometry (TFI) label-free immunoassay analyzer (Gator Bio, Palo Alto, CA). Each serum sample was diluted in a series (1:25, 1:100, 1:250, 1:500, 1:1000, 1:2000) in running buffer (PBS at pH 7.4 with 0.02% Tween-20, 0.2% BSA, and 0.05% $NaN_3$) for analysis. The first testing cycle for each diluted sample measured the binding ability of RBD to ACE2 with neutralization, and the second cycle provided the full binding ability of RBD without neutralization. In each cycle, the recorded time course of signals, as known as the sensorgram, was recorded. The readout measured the signal increase in RBD-ACE2 complex formation, representing the quantity of RBD-ACE2 complex on the sensing probe. A neutralization rate was calculated as the ratio of the readout in the first cycle to that in the second cycle, presented as a percentage. To obtain the neutralizing antibody titer ($IC_{50}$) for each serum sample, the neutralization rates were plotted against dilutions, and the points were fitted using a linear

interpolation model. The reciprocal of the dilution resulting in a 50% neutralization rate was defined as the neutralizing antibody titer.

## SARS-CoV-2 Spike protein western blot

Maternal blood and cord blood were diluted in RNAlater in 1:1.3 ratio, placenta was preserved in RNAlater. Protein lysates were obtained from samples using RIPA buffer (150 mM NaCl, 25 mM Tris-HCl (pH 7.4), 1% NP-40, 0.5% sodium deoxycholate, 0.1% sodium dodecyl sulfate) containing Halt™ protease inhibitor cocktail (ThermoScientific). Cell Lysates were resolved by SDS/PAGE on a Bis-Tris methane 4–12% poly-acrylamide gel and transferred to a nitrocellulose membrane, blocked with 5% skimmed milk diluted in PBS, an incubated overnight at 4 °C with anti-SARS-CoV-2 Spike mouse mAb (1A9, GeneTex) or anti-GAPDH rabbit polyclonal antibody (GTX100118, GeneTex), respectively, diluted 1:1000 or 1:5000 in blocking buffer. The membrane was washed in PBS buffer containing Tween-20 (0.1%) and then incubated for 1 h with horseradish peroxidase-conjugated anti-mouse and anti-rabbit secondary antibody (Jackson ImmunoResearch) diluted, respectively, 1:5000 and 1:10,000. The membrane was thoroughly washed, and proteins visualized using Immobilon Forte Western HRP substrate (Millipore). Samples were compared to a positive control of 293T cells constitutively expressing wild-type SARS-CoV-2 Spike protein. Dynamic range of the standard curve of the Spike protein western blot was 5 ng to 200 ng (Supplementary Fig. 2).

## SARS-CoV-2 Spike mRNA PCR

Maternal blood and cord blood were diluted in RNAlater in 1:1.3 ratio, placenta was preserved in RNAlater. Tissues were kept at −80 °C until analyzed. RNA was isolated from samples using the RNeasy Micro or Mini Kit (Qiagen) according to manufacturer's protocol. RNA concentration was measured using nanodrop and all samples had >30 ng/µL total RNA. 500 ng RNA was transcribed into cDNA using qScript cDNA synthesis kit (Quantabio). Primers were design to detect the vaccines mRNA (mRNA-1273 Moderna and BNT162b2 Pfizer-BioNtech) as previously described[61]. Forward primer: AACGCCACCAACGTGGT-CATC. Reverse primer: GTTGTTGGCGCTGCTGTACAC. Primers were shown to detect samples containing as low as 1.5 pg of vaccine using vaccine standard curve (Table S2). QuantaStudio 6 Flex (Applied Bio-systems) instrument and SsoFast EvaGreen supermix (Bio-Rad) were used for PCR reaction: 30 s 95 °C followed by 40 cycles of 5 s 95 °C and 20 s 60 °C. All samples were run in triplicate as 20 µL reactions, and Ct values corresponding to <1.5 pg of vaccine based on standard curve (Table S2) were interpreted as a negative result. For vaccines cDNA standard curves, 10,000 pg/µL vaccine mRNA (as cDNA) sample was used for serial dilution in 1:3 ratio, up to 0.06 pg/µL. Two micro-liters of these diluted samples were used in each well to create standard curves.

## PhIP-Seq/VirScan Coronavirus phage display assay

**Immunoprecipitation of phage-bound patient antibodies.** Maternal plasma at delivery and cord plasma were evaluated by PhIP-Seq/Virscan Coronavirus phage display. Construction of the Coronavirus PhIP-Seq library and detailed methods for immunoprecipitation, sequencing, and bioinformatic processing of data are identical to what has previously been described[28]. For the purposes of the analysis conducted in this study, analysis was restricted to sero-reactivity against the SARS-CoV-2 Spike protein. As previously described, a total of two rounds of amplification and selection were performed for all PhIP-Seq analyses.

**Next-generation sequencing library prep.** Amplicon sequencing library preps were performed using the Labcyte Echo 525 and an Integra Via Flow 96 and were identical to what has previously been described[28]. All libraries were pooled by equal volume, cleaned and size selected using Ampure XP beads at 1.0X per manufacturer's protocol. Libraries were quantified by High Sensitivity DNA Qubit

and quality-checked by High Sensitivity DNA Bioanalyzer. Sequencing was then performed on a NovaSeq S1 (300 cycle kit with 1.3 billion clusters) aiming for sequencing depths of at least 1 million reads per sample.

**Bioinformatic analysis of PhIP-Seq data.** Sequencing reads were aligned to a reference database of the full viral peptide library using the Bowtie2 aligner. For all VirScan libraries, the null distribution of each peptide's log10(rpK) was modeled using a set of 95 pre-pandemic, healthy control sera. All counts were augmented by 1 to avoid zero counts in the healthy control sera samples. Multiple distribution fits were examined for these data, with the normal distribution showing the best fit. These null distributions were used to calculate $p$ values for the observed log10(rpK) of each peptide within a given sample. The calculated $p$ values were corrected for multiple hypothesis using the Benjamini–Hochberg method. Any peptide with a corrected $p$ value of <0.001 was considered significantly enriched over the healthy background. To identify regions targeted by host antibodies, all library peptides were aligned to the SARS-CoV-2 reference genome. Using the aligned position of the significantly enriched peptides which aligned full-length against the reference, we determined the proportion of individuals (mothers and infants) that were reactive at each residue of the Spike protein. All plots were generated using the R ggplot2 package.

## Statistical analysis

Statistical analyses were performed using PRISM v9.2 (GraphPad), STATA 16 (StataCorp), and R version 3.6.3 and R Studio version 1.1.447. Descriptive statistics include mean, standard deviations, and ranges for continuous variables. The Wilcoxon rank-sum test was used for two-group comparisons of continuous variables including maternal pre- and post-vaccine antibody responses. Associations between continuous variables were assessed using Spearman's rank correlation ($R_s$) including comparisons between maternal, cord and infant follow-up antibody IgG and neutralizing titer responses, and timing of vaccination. Two-sided $p$ values were calculated for all test statistics, and $p < 0.05$ was considered significant. PhIP-Seq/VirScan bioinformatics as detailed above.

## Reporting summary

Further information on research design is available in the Nature Research Reporting Summary linked to this article.

# Data availability

The PhIP-Seq/VirScan data generated in this study have been deposited in the dryad database under accession code: https://doi.org/10.7272/Q6DJ5CWD. Source data are provided with this paper.

# Code availability

Code generated during and/or analyzed during the current study are available from the corresponding author on reasonable request.

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

## Acknowledgements

M.P. was supported by the National Institutes of Health (NIAID K23AI127886), the Marino Family Foundation, and UCSF REAC award. Y.G. was supported by the Weizmann Institute of Science - National Postdoctoral Award Program for Advancing Women in Science, and of the International Society for Research in Human Milk and Lactation (ISRHML) Trainee Bridge Fund. Y.M. and W.C.G. were supported by The Roddenberry Foundation. S.L.G. was supported by the National Institutes of Health (NIAID K08AI141728) and a generous gift from the Kryzewski Family. We thank all the mothers and infants that participated in this study. We thank Kenneth Scott, BS, RPh (UCSF Health Pharmacy, University of California, San Francisco) and Hannah J. Jang, PhD, RN, PHN, CNL (UCSF School of Nursing, University of California, San Francisco), for voluntarily providing unused vaccine for this study, and to Dr. Margaret Feeney (University of California, San Francisco) and Dr. Nadav Ahituv (University of California, San Francisco) for support of these experiments.

## Author contributions

M.P. helped conceive and design the project, oversaw recruitment, designed, and performed sample collection, oversaw experiment design, oversaw data analysis, provided funding, and drafted the manuscript. Y.G. Recruited and consented enrollees, oversaw sample collection, designed, performed, and analyzed mRNA PCR experiments, performed data analysis. A.G.C. Recruited and consented enrollees, oversaw sample collection, performed chart review, and helped draft the manuscript. Y.M. Performed and helped design western blot assay. L.L. Performed and analyzed mRNA PCR experiments, performed sample collection. B.A. Performed phage immunoprecipitation assays. H.C. and U.J. performed and helped design critical experiments, and data collection. C.Y.L., V.J.L., M.C., L.W., and S.B. Performed and coordinated sample collection, and data collection. V.J.F. Helped conceive and coordinate the project. A.P.M. Provided funding. W.C.G. Helped design western blot and oversaw data analysis. A.H.B.W. Designed and oversaw all serology experiments. K.L.L. Designed and oversaw all neutralizing antibody experiments. J.R. Designed, analyzed, and oversaw phage immunoprecipitation sequencing assays. S.L.G. conceived and designed the project, oversaw recruitment, oversaw experiment design, oversaw data analysis, provided funding, and helped draft the manuscript. M.P., Y.G., Y.M., L.L., A.H.B.W., W.C.G., K.L.L., and S.L.G. verified the underlying data.

## Competing interests

The authors declare no competing interests. All authors reviewed and approved the manuscript.
