## [Peer Review File · Nature Communications]

REVIEWER COMMENTS

Reviewer #1 (Remarks to the Author):

In this study, Prah et al. evaluate the transplacental transfer of mRNA vaccines and anti-SARS-CoV-2 antibodies during pregnancy in a cohort of 20 individuals vaccinated during pregnancy (8 Pfizer, 12 Moderna) – of these patients, 18 had received both vaccine doses prior to pregnancy and 2 received the 2nd dose after delivery. They find no evidence of mRNA vaccine components in the maternal blood, placenta, or infant cord blood. There was transplacental transfer of SARS-CoV-2 IgG and neutralizing antibodies into the cord blood that persisted in the newborns. Additionally, a vaccine-specific Spike protein epitope binding signature was identified using phage immunoprecipitation sequencing.

SUMMARY:

In this work, the authors aim to address several very important and clinically-relevant questions regarding vaccination with the mRNA COVID-19 vaccines in pregnancy: (1) does the vaccine cross the placenta?, (2) what are the characteristics of the maternal and cord COVID-19-vaccine induced antibodies? The work presented is limited by the small sample size of pregnant individuals assessed and by the methods used to address these questions. Specifically, only delivery samples were assessed for mRNA vaccine components, which, for most subjects, was many days to weeks after completion of vaccination. As the vaccine mRNA and subsequent translated Spike protein would only be anticipated to be present for a very short time (i.e., hours to days) following vaccination, one would not expect to find mRNA vaccine components in samples obtained weeks later. A more durable indication of transplacental mRNA vaccine transfer would be the presence of anti-SARS-CoV-2 Spike IgM antibodies in the cord blood. However, I have concerns that the IgM ELISA utilized in these experiments did not perform as anticipated – the maternal plasma samples should have had a robust IgM signature following vaccination (as demonstrated by others, including in PMID: 33775692). While the authors here see the expected maternal IgG response following vaccination (Fig 1A), they do not observe the expected IgM response. Thus, I have questions about the ability of their utilized methods to detect IgM in the cord blood samples if it was present.

Additional concerns/ comments are detailed below:

1. Detection of mRNA vaccine:

- Components of the mRNA vaccine would be anticipated to be found in the maternal plasma hours to days following vaccination (not weeks) – these are the timepoints that should be assessed. In review of Table S1, there were no participants who delivered within 2 days of vaccine dose #1 (shortest was 9 days).

- In previous work, others have demonstrated that the S1 antigen is present as early as day 1 post vaccination, with peak levels on day 5 post vaccination, and undetectable by day 14 (PMID: 34015087). Nucleocapsid was used as a negative control. Detection of the S1 antigen was only possible using an ultra sensitive single molecular array (Simoa)—this type of assay has ultra low detection limits allowing for the assessment of the extremely low levels of antigen present following vaccination. S1 was only detectable following the first vaccine dose, and was not detected following the 2nd dose. Western blot is an extremely insensitive technique to detect the presence of proteins and would not be anticipated to be able to detect the very low (i.e., subpicomolar) concentrations of antigens present following vaccination.

- Thus the timepoints chosen for assessment and methodologies used are inadequate to assess for mRNA vaccine products in these pregnant individuals

2. Maternal antibody response to mRNA vaccination

- In the methods for assessment of SARS-CoV-2 plasma serology, the authors write that they assess antibodies against RBD and nucleocapsid protein (NP). In the results, IgG and IgM levels are reported as anti-SARS-CoV antibodies without specification of RBD or NP. What specific antibodies are displayed in Figure 1? As only RBD serologies (not NP) should increase following COVID-19 vaccination, did any subjects have positive NP antibodies suggestive of prior SARS-CoV-2 infection? Is there a reason that Spike IgM and IgG were not assessed? This may have given a more robust assessment of IgM response following vaccination in review of other literature.

- Given the lack of expected IgM response in the maternal plasma, I have concerns about the reliability of the assessment for IgM in the cord blood

- Similar to Figure 1, are the anti-SARS-CoV-2 IgG's in Figure 2 (A, C, E), Figure 3, and Figure 4 (A, B, E, F) IgG's against RBD?

3. Transplacental transfer of anti-SARS-CoV-2 IgG antibodies

- The authors comment that no anti-SARS-CoV-2 IgM was detected in any of the cord blood samples. What were the levels of maternal IgM at the time of delivery? While these wouldn't be transplacentally transferred, they would give an indication of decay of IgM levels since time of vaccination. (see above re: concerns for assessment of IgM levels)

4. Persistence of maternally-derived SARS-CoV-2 IgG antibodies in neonatal blood

- Of the infants (n=11) with blood antibody levels assessed between 3-15 weeks of life, 9 of 11 had anti-SARS-CoV-2 IgG detected. Was this RBD IgG?

- When neutralizing titers were assessed in these 9 of 11 infants with IgG, only 8 babies had data (not 9) (line 142). Why is this the case?

5. Timing of vaccination and maternal antibody levels

- The authors state that they find no correlation between timing of vaccination and maternal antibody levels at delivery (lines 160-161). However, if the 2 subjects without adequate time since vaccination are removed, it appears that time since vaccination and maternal antibody levels would be correlated (which would be expected) (Fig 4A). This data should be re-analyzed without the outliers.

Other points:

- In the methods for sample collection, the methods suggest that all whole blood collected was placed immediately in RNAlater. Was this done only for the samples on which mRNA vaccine components were to be collected? Were samples for SARS-CoV-2 serologies just plasma samples isolated from whole blood? Were the tube types for maternal, cord, and infant blood collection similar? This should be clarified in this section.

- In the methods for SARS-CoV-2 plasma serology, please clarify why only RBD and NP were assessed (and not Spike) and how this translates to what is depicted in the Figures.

- In the methods for the Western blot, the sensitivity of this technique for detection of Spike protein should be stated and a control with different concentrations of Spike protein should be present on the Western blot. As stated above, this technique would not be anticipated to be sensitive enough to detect the very low amounts of Spike protein present in the maternal circulation in the days directly following vaccine dose 1.

Reviewer #2 (Remarks to the Author):

The article "Evaluation of transplacental transfer of mRNA vaccine products and functional antibodies during pregnancy and early infancy » reports preliminary interesting results concerning efficient passive transfer of a protective immunity to neonates born to mothers vaccinated by mRNA vaccine during pregnancy.

There are several main points to address:

Overall, the initial cohort of 20 pregnant women does not allow having full information expected, as most subgroups are much smaller due to lack of seroconversion in the mother and missing samples in the neonates (especially for the first weeks follow-up). For example lines 132, 142, 145... This point makes it

more difficult to drive any formal conclusions and authors should discuss this point and underline the limits of their study.

Concerning maternal serology (lines 99-101 and 303), it is not clear whether anti-N and anti-S are detected separately. Moreover, only four patients had serology before vaccination (figure 1); can authors provide clear information on the absence of prior exposure to the virus for the other patients also?

Lines 218-222: authors rely on the absence of IgM to conclude that there is no transfer of vaccine product. However, IgM are commonly unreliable to diagnose a congenital infection (Kimberlin, JAMA, 2020. Voordouw, Clinical Microbiology Reviews, 2020. Schwartz, Archives of Pathology and Laboratory Medicine, 2020). Please discuss this point.

Line 223 and 281-282: vaccine products could be safe even if transmitted to the fetus. On the other hand, even if unlikely, vaccine could induce inflammatory response in the mother that could have negative impact on the neonate (Gee, Nat Immunol 2021). Authors don't have enough patients and background that allows driving this conclusion.

Reviewer #3 (Remarks to the Author):

This manuscript reports on 20 people who received an mRNA vaccination during pregnancy, and 19 infants born subsequently. There are four main findings.

1. Following vaccination, IgM is not detected in cord blood. This is in line with previous findings (Mithal, 2021, AJOG; Beharier, 2021, JCI) which should be cited. The authors show, for the first time, that spike mRNA and protein is not detectable in placenta or cord blood. The authors use these findings to argue that the vaccine does not cross the placenta. Although this is likely to be the case, they will make a stronger argument if they consider the kinetics of vaccine biodistribution and the IgM response. Biodistribution studies (eg. <https://byrambridle.com/docs/bio-dist-eng.pdf> for BNT162b2) show vaccine clearance within approximately a week, and IgM generally declines below the threshold of detection approximately six weeks after immunisation. The authors should focus on those participants who received either their first or their second dose in a timeframe such that vaccine mRNA/protein or IgM would still be detectable if they were present.

2. Vaccine-specific IgG, including neutralising antibodies, are transferred to the infant. This is already well-established - see for example (Gray, AJOG, 2021; Collier, JAMA, 2021; Rottenstreich, CMI, 2021). In particular, it should be noted that a very granular analysis of how timing of vaccination in pregnancy affects umbilical cord blood IgG titres has recently been published (Yang, Obs Gynecol, 2021), which should be cited. The authors claim that functional neutralizing antibodies are differentially transferred to the fetus (Line 177). To assess this claim, we would need to see a comparison between paired transfer ratios for total IgG vs neutralising antibody, and I cannot see that this has been done. The authors should either do this analysis, or remove the claim.

3. mRNA vaccination leads to a unique SARS-CoV2 Spike protein antibody epitope binding signature, with some of the targeted regions unique to vaccination. However, all the participants in this study were pregnant, whereas none of the participants in the three cited studies on epitope binding following SARS-CoV2 infection were. Although I agree it seems more likely that the key difference is infection versus vaccination, the authors cannot rule out the possibility that the key difference in antibody raised is in fact pregnancy (see for example Atyeo, 2021, Sci Transl Med, who report a subtle difference in the antibody response to the first dose of vaccination during pregnancy). To make this conclusion more firmly, non-pregnant participants should be examined following vaccination. At the very least, this issue should be discussed.

4. “We believe we are the first to report that infants in the first few months of life continued to have maternal vaccine-derived anti-SARS-CoV-2 antibodies.” Yes, although it’s worth noting that Shook et al have also preprinted this on MedRxiv – this should be cited.

Finally, the authors make a statement to the effect of “timing of vaccination during pregnancy is critical to ensure transplacental transfer of protective antibodies during early infancy” a number of times. This is at odds with the rationale for recommending COVID vaccination during pregnancy, which is to protect mother and baby *during* pregnancy. Suggesting that there is an ideal time in pregnancy to get vaccinated, for the benefit of the infant after it is born, could be counterproductive, in that it may encourage people who are willing to get vaccinated to delay their vaccination, to maximise protection of the newborn. I would suggest changing the emphasis here.

Response to Reviewers:

Reviewer #1 (Remarks to the Author):

In this study, Prah et al. evaluate the transplacental transfer of mRNA vaccines and anti-SARS-CoV-2 antibodies during pregnancy in a cohort of 20 individuals vaccinated during pregnancy (8 Pfizer, 12 Moderna) – of these patients, 18 had received both vaccine doses prior to pregnancy and 2 received the 2nd dose after delivery. They find no evidence of mRNA vaccine components in the maternal blood, placenta, or infant cord blood. There was transplacental transfer of SARS-CoV-2 IgG and neutralizing antibodies into the cord blood that persisted in the newborns. Additionally, a vaccine-specific Spike protein epitope binding signature was identified using phage immunoprecipitation sequencing.

SUMMARY:

In this work, the authors aim to address several very important and clinically-relevant questions regarding vaccination with the mRNA COVID-19 vaccines in pregnancy: (1) does the vaccine cross the placenta? (2) what are the characteristics of the maternal and cord COVID-19-vaccine induced antibodies? The work presented is limited by the small sample size of pregnant individuals assessed and by the methods used to address these questions. Specifically, only delivery samples were assessed for mRNA vaccine components, which, for most subjects, was many days to weeks after completion of vaccination. As the vaccine mRNA and subsequent translated Spike protein would only be anticipated to be present for a very short time (i.e., hours to days) following vaccination, one would not expect to find mRNA vaccine components in samples obtained weeks later. A more durable indication of transplacental mRNA vaccine transfer would be the presence of anti-SARS-CoV-2 Spike IgM antibodies in the cord blood. However, I have concerns that the IgM ELISA utilized in these experiments did not perform as anticipated – the maternal plasma samples should have had a robust IgM signature following vaccination (as demonstrated by others, including in PMID: 33775692). While the authors here see the expected maternal IgG response following vaccination (Fig 1A), they do not observe the expected IgM response. Thus, I have questions about the ability of their utilized methods to detect IgM in the cord blood samples if it was present.

We thank the reviewers for this important point. We agree that the performance of IgM assay to detect cord blood immune responses to SARS-CoV-2 antigen was key to this study. Thus, we used a clinically validated anti-SARS-CoV-2 IgM and IgG assay for which we have extensive experience in detecting IgM responses to SARS-CoV-2 infection and SARS-CoV-2 vaccination in our prior publications (PMID: 33501951, PMID: 34804068, and PMID: 34234001). We have updated citations in lines 69, 252-254, and 350-353 with references to this work. Additionally, prior work by our co-author Dr. Alan Wu has determined the specificity of our anti-SARS-CoV-2 IgM assay to be 100% and sensitivity by post-infection day as follows: Days 1-5 19%, Days 6-10 51%, days 11-15 64%, days 16-20 days 61%, and 21-28 was 93% (<https://www.testmenu.com/zsfglab/Tests/1038493>). As referenced in supplementary table 1, 18/20 of our samples at delivery are greater than 21 days since vaccine dose 1, and 2/20 samples at delivery are within 6-10 days of vaccine dose 1, which would fit in the timeframe of 93% and 51% assay sensitivity, respectively.

Most importantly, we have previous experience with this anti-SARS-CoV-2 IgM assay to detect a fetal anti-SARS-CoV-2 IgM response, as we have previously used the same assay in a cohort of maternally infected COVID-19 mother-infant dyads (PMID: 34234001). We were able to detect a vertical viral transmission of SARS-CoV-2 from mother to baby. The infant was born at 31 weeks of gestation after an asymptomatic maternal infection of which the mother was

diagnosed on the day of premature delivery. The vertical transmission was confirmed by SARS-CoV-2 PCR positivity in maternal blood at delivery, infant meconium, and infant nasopharyngeal swab (positive at 24 hours of life and was persistently positive throughout hospitalization). The premature infant was IgM and IgG negative in cord blood, but the infant became IgM and IgG positive on the next blood sampling evaluation at day of life 16 (no blood was taken prior to this date), and the infant continued to be IgM positive at 8 weeks of life. This infant was premature, and the mother's SARS-CoV-2 infection was only detected on the day of delivery, and the subsequent infant serologic response fit the known kinetics of SARS-CoV-2 IgM responses. Thus, in the case of a fetal SARS-CoV-2 antigen exposure weeks prior to delivery, we would expect to see an IgM response by the infant using this assay. Additionally, no infant had a rising IgG level above cord blood levels at the time of infant follow up (Fig 2C, D), which in our prior study we found our infant with vertical transmission had a rising IgG levels at follow up time points.

Lastly, we appreciate the reference to publication (Gray KJ, et al. PMID: 33775692) which evaluated IgM responses to mRNA vaccination in pregnancy. Though, it is difficult to directly compare the IgM assay in the referenced paper to our IgM assay, as our assay was clinically validated to determine the positive cutoff value for a high level of sensitivity and specificity. The referenced paper (Gray, KJ et al.) found all participants had detectable IgM levels at post-vaccination times points, however they also had detected levels of IgM above media (PBS) background in all pre-vaccine samples (Suppl fig 1 PMID: 33775692) raising concerns about the specificity of that referenced assay.

Additional concerns/ comments are detailed below:

1. Detection of mRNA vaccine:

- Components of the mRNA vaccine would be anticipated to be found in the maternal plasma hours to days following vaccination (not weeks) – these are the timepoints that should be assessed. In review of Table S1, there were no participants who delivered within 2 days of vaccine dose #1 (shortest was 9 days).
- In previous work, others have demonstrated that the S1 antigen is present as early as day 1 post vaccination, with peak levels on day 5 post vaccination, and undetectable by day 14 (PMID: 34015087). Nucleocapsid was used as a negative control. Detection of the S1 antigen was only possible using an ultra-sensitive single molecular array (Simoa)—this type of assay has ultra-low detection limits allowing for the assessment of the extremely low levels of antigen present following vaccination. S1 was only detectable following the first vaccine dose, and was not detected following the 2nd dose. Western blot is an extremely insensitive technique to detect the presence of proteins and would not be anticipated to be able to detect the very low (i.e., subpicomolar) concentrations of antigens present following vaccination.
- Thus the timepoints chosen for assessment and methodologies used are inadequate to assess for mRNA vaccine products in these pregnant individuals

We have updated supplementary table 1 to clarify the timing of the two vaccine doses from delivery per participant. The range of time from vaccine doses is as follows: time from vaccine dose 1 ranged from 6-97 days prior to delivery, time from vaccine dose 2 ranged from 2-75 days prior to delivery, and in two participants 15 and 21 days after delivery. We were unable to obtain delivery specimens from participant 11085 (delivered at an outside hospital). Thus, we evaluated samples for the detection of mRNA vaccine products for timepoints that ranged from 9-97 days from vaccine dose 1 from delivery, and 2-75 days from vaccine dose 2.

We thank the reviewer for this important point. Concerns by anti-vaccine proponents have attempted to raise concerns about the possibility of vaccines crossing the placenta and reaching the fetus as an argument against vaccination in pregnancy. Thus, further data is needed to evaluate safety and provide confidence in evidence to support vaccination in pregnancy.

We agree that the detection of mRNA vaccine products in humans after vaccination is challenging, with prior studies often with small sample sizes and evaluation at varying timepoints from vaccination. However, we believe that further studies are needed to evaluate potential clinically relevant levels of vaccine products including in different physiologic states (including pregnancy) and the potential of tissue persistence (placenta) or transplacental transfer after vaccination.

To evaluate the detection of clinically relevant levels of mRNA vaccine products after vaccination in pregnancy, we used multiple methods to evaluate potential transplacental transfer of mRNA vaccine products: mRNA qPCR, Spike protein Western blot, and fetal IgM production. We first assessed for the presence of vaccine mRNA and Spike protein in maternal blood, placenta, and cord blood. Our first method was mRNA qPCR in delivery specimens which had a sensitivity of detection to 1.5 pg/uL (supplemental table 2). Then to evaluate the potential of Spike antigen presence we used a Spike antigen Western blot. We have since completed additional experiments to determine the dynamic range of sensitivity our S1 protein Western blot. Western blot was performed using purified S1 protein and the signal intensity of the bands was quantified using a densitometer. S1 protein was undetectable at 2.5 ng, and the signal intensity became not linear over 200 ng; the dynamic range of the standard curve was 5 ng to 200 ng. We have updated our methods in lines 402-405 and added supplementary figure 2 with this information.

We thank the reviewer for the reference (Ogata, AF et al. PMID: 34015087) which evaluated circulating SARS-CoV-2 mRNA vaccine antigens to mRNA-1273 S1 and Spike protein in plasma in a small study of 13 participants after mRNA vaccination. S1 protein was detected in plasma beginning on day 0 post-vaccination dose 1 through day 9, and then day 29 (2 days following dose 2). Spike protein was detected up to day 28 after one dose and day 29 after dose one (1 day after dose 2). Limitations of this study include lack of evaluation of tissue, they did not evaluate Pfizer vaccine.

Interestingly, since submission, a small study by Santos A et al (PMID: 35361888) evaluated placenta tissue for mRNA vaccine related Spike protein by immunohistochemistry after maternal vaccination (ranging from 2-79 days after last dose) and did not find evidence of mRNA vaccine products after maternal vaccination, consistent with our conclusions. Immunohistochemistry is generally more sensitive than Western blot for detection of protein, but Western blot analysis is more consistent for protein quantification from given input of cells/tissue. The lack of detection down to 5 ng in our immunoblot assay provides additional evidence that mRNA vaccine products are not significantly transferred to the placenta. We have updated this citation in line 238.

Additionally, since submission Roltgen et al. (PMID: 35148837) evaluated detection of vaccine mRNA in local lymph nodes after mRNA-1273 and BNT162b2 vaccination (RNAscope for mRNA vaccine, immunohistochemical staining for spike antigen) and plasma (antigen capture ECL immunoassay platform) in a small study (n=7). Vaccine mRNA was detected in local lymph nodes up to day 60 post dose 2, mRNA and Spike antigen levels were not quantified in lymph node tissue and thus is it difficult to compare to our assays. In plasma, Spike protein was

detected at day 1-2 in most (median concentration was 47 pg/mL and in ~63% of participants day 7 post dose 1. After vaccine dose 2 and half positive within 1-2 days with one participant positive 7 days after dose 2.

Another recent study by Yeo KT et al PMID: 35087517 evaluating the persistence of vaccine mRNA by PCR in serum and breastmilk found 40% of BNT162b2 vaccinated mothers had detectable vaccine mRNA 7-10 days after dose 2 (27-31 days post dose 1) with median levels 12 ng/100mL in serum.

We have updated lines 238-241 to highlight four recent studies from Ogata, AF et al. (PMID: 34015087), Roltgen K et al. (PMID: 35148837), Yeo KT et al (PMID: 35087517), and Santos A et al (PMID: 35361888).

The majority of our participants had specimens evaluated for mRNA vaccine products at delivery that were within the range of positive detection of mRNA products of 60 days from dose 2 as seen in the Roltgen et al. study. Though, they evaluated the presence of mRNA vaccine products in local lymph nodes, and we evaluated placental tissue and maternal and cord blood at delivery, which directly investigates concerns about maternal-fetal or placental effects of the vaccine in the pregnant population. Further studies are needed to evaluate potential persistence of vaccine products including in different physiologic states (including pregnancy) and the potential of tissue persistence in distal sites (placenta), and the potential for cross-over to the fetus.

We acknowledge the limitations of quantification of potential vaccine products in our presented assays. However, the goal of our study was not to detect any potential vaccine product at subpicomolar levels, it was to detect the potential for mRNA vaccine products to cross over to the fetus at clinically relevant levels. Thus, our final approach to determine the potential of transplacental transfer of vaccine antigen was to use fetal IgM production to determine the possibility if any clinically relevant levels vaccine antigen had been transplacentally transferred. If vaccine antigen had reached the fetus at levels below the limit of detection of our assays but at physiologically relevant levels, we would have expected an endogenous infant immune response (IgM detection and/or rising IgG levels in the infant) to the *in utero* antigen exposure, which did not find.

2. Maternal antibody response to mRNA vaccination

- In the methods for assessment of SARS-CoV-2 plasma serology, the authors write that they assess antibodies against RBD and nucleocapsid protein (NP). In the results, IgG and IgM levels are reported as anti-SARS-CoV antibodies without specification of RBD or NP. What specific antibodies are displayed in Figure 1? As only RBD serologies (not NP) should increase following COVID-19 vaccination, did any subjects have positive NP antibodies suggestive of prior SARS-CoV-2 infection? Is there a reason that Spike IgM and IgG were not assessed? This may have given a more robust assessment of IgM response following vaccination in review of other literature.
- Given the lack of expected IgM response in the maternal plasma, I have concerns about the reliability of the assessment for IgM in the cord blood
- Similar to Figure 1, are the anti-SARS-CoV-2 IgG's in Figure 2 (A, C, E), Figure 3, and Figure 4 (A, B, E, F) IgG's against RBD?

For this study we used an assay that had a well-validated sensitivity and specificity for both anti-SARS-CoV-2 IgM and IgG, and we used the same assay as our prior publications for reasons stated above. This assay detects anti-SARS-CoV-2 antibody to RBD and nucleocapsid in a combined assay. We have further clarified the combined assay in the methods (lines 353-355). Though no participants reported a history of known COVID-19 infection in our enrollment survey (updated lines 339-340).

To ensure all reported antibody responses were only to SARS-CoV-2 RBD (and not nucleocapsid), we have completed additional experiments to evaluate for anti-nucleocapsid antibody responses in maternal plasma collected at delivery for which all participants were negative (updated in lines 97-98 and 364-368). We have updated all figure legends (lines 520, 522, 547, 552, 555, 574, 596, 597, 604, 606, 607) to clarify the use of the combined anti-SARS-CoV-2 to RBD and nucleocapsid assay.

3. Transplacental transfer of anti-SARS-CoV-2 IgG antibodies

- The authors comment that no anti-SARS-CoV-2 IgM was detected in any of the cord blood samples. What were the levels of maternal IgM at the time of delivery? While these wouldn't be transplacentally transferred, they would give an indication of decay of IgM levels since time of vaccination. (see above re: concerns for assessment of IgM levels).

We have added an additional figure of maternal IgM at delivery by time since vaccine dose 1 (Fig 4A, text line 176-178). In this study we evaluated serologic responses 3-4 weeks following vaccine dose one, and 4-8 weeks following vaccine dose two. Due to the limitations of our timepoints evaluated, we were not able to assess the beginning of the rise of IgM. Regarding the persistence of IgM, we found one sample collected post-vaccine dose 2 still positive for IgM 69 days after dose 1 and 43 days after dose 2. Limited data exists on IgM responses to vaccine doses, how they differ from natural infection and if a 2nd dose of vaccine impacts IgM trajectory of responses. However, our co-authors (Lynch KL et al PMID: 33501951) have previously reported the IgM trajectory with this assay to SARS-CoV-2 infection with median time to IgM seroconversion day 10 (IQR 7-14). Additionally, using this same IgG and IgM assay in a pregnancy cohort after SARS-CoV-2 infection (Song D et al. PMID: 34234001), we found IgM to be positive out to 98 days from SARS-CoV-2 PCR diagnosis. We evaluated cord blood from mothers vaccinated up to 6-97 days prior to delivery, which would be within the range of potential IgM positivity in prior assessments of this IgM assay.

4. Persistence of maternally-derived SARS-CoV-2 IgG antibodies in neonatal blood

- Of the infants (n=11) with blood antibody levels assessed between 3-15 weeks of life, 9 of 11 had anti-SARS-CoV-2 IgG detected. Was this RBD IgG?
- When neutralizing titers were assessed in these 9 of 11 infants with IgG, only 8 babies had data (not 9) (line 142). Why is this the case?

All IgG and IgM from maternal, cord, and infant blood was to the combined anti-SARS-CoV-2 RBD/N antibody assay as described in methods lines 97-98 and 364-368. We have updated the figure legends lines 520, 522, 547, 552, 555, 574, 596, 597, 604, 606, 607 to clarify the use of the combined anti-SARS-CoV-2 RBD/N antibody assay.

We attempted neutralizing assays on all 9 infants with positive IgG levels at follow up. Unfortunately, we were unable to determine the neutralizing titer in one of the infant samples due to an interfering substance, and due to the sample volume of the infant samples, we did not have an additional sample to re-evaluate for the assay.

5. Timing of vaccination and maternal antibody levels

- The authors state that they find no correlation between timing of vaccination and maternal antibody levels at delivery (lines 160-161). However, if the 2 subjects without adequate time since vaccination are removed, it appears that time since vaccination and maternal antibody levels would be correlated (which would be expected) (Fig 4A). This data should be re-analyzed without the outliers.

We thank the reviewer for this comment and agree. We have updated our analysis and removed the two participants who received the first dose of vaccine <30 days prior to delivery (lines 174-184). As expected, there is now a statistically significant association between maternal delivery IgM, IgG, and neutralizing antibody levels and time since vaccine dose 1.

Since one of our main conclusions focuses on how the timing of vaccine administration well prior to delivery was needed in order for antibodies to rise to detectable levels at the time of delivery and transplacental transfer, we still wanted to visually display how vaccine administration too close to delivery (<30 days prior to delivery) was associated with low or absent levels at delivery. Thus, we updated the statistical analysis in Figure 4A-E to be consistent with the updated analysis (lines 174-184), but we did maintain the data points on the graph for participants that received their vaccine dose one <30 days prior to delivery, but we demarcated them separately (half circles)— and did not include those participants in the presented statistical analysis. If the reviewer recommends removal of these participants, we would be happy to update the graphs to only analyzed data points.

Other points:

- In the methods for sample collection, the methods suggest that all whole blood collected was placed immediately in RNAlater. Was this done only for the samples on which mRNA vaccine components were to be collected? Were samples for SARS-CoV-2 serologies just plasma samples isolated from whole blood? Were the tube types for maternal, cord, and infant blood collection similar? This should be clarified in this section.

We have updated lines 343-344 with further information on blood processing. All maternal, cord, and infant blood was collected in EDTA tubes and processed with the same workflow. All assays reported in the manuscript including evaluation of mRNA products and serologies were completed from same collected blood sampling tubes.

- In the methods for SARS-CoV-2 plasma serology, please clarify why only RBD and NP were assessed (and not Spike) and how this translates to what is depicted in the Figures.

We used a well-validated anti-SARS-CoV-2 antibody assay that we have previously shown to detect IgM and IgG antibody levels after maternal infection, fetal vertical transmission, and vaccination in lactation, and we used the anti-SARS-CoV-2 combined RBD/nucleocapsid combined immunoassay (see above). All presented serologic results and figures are from the combined Anti-SARS-CoV-2 RBD and N protein assay (methods line lines 97-98 and 364-368). We have updated figure legends lines 520, 522, 547, 552, 555, 574, 596, 597, 604, 606, and 607 to reflect this.

- In the methods for the Western blot, the sensitivity of this technique for detection of Spike protein should be stated and a control with different concentrations of Spike protein should be present on the Western blot. As stated above, this technique would not be anticipated to be

sensitive enough to detect the very low amounts of Spike protein present in the maternal circulation in the days directly following vaccine dose 1.

We have completed additional experiments to determine the dynamic range of our S1 Western Blot which ranged from 5 ng to 200 ng. We have added supplementary figure 2 with the Spike protein Western blot standard curve and lines 403-405.

Reviewer #2 (Remarks to the Author):

The article "Evaluation of transplacental transfer of mRNA vaccine products and functional antibodies during pregnancy and early infancy » reports preliminary interesting results concerning efficient passive transfer of a protective immunity to neonates born to mothers vaccinated by mRNA vaccine during pregnancy.

There are several main points to address:

Overall, the initial cohort of 20 pregnant women does not allow having full information expected, as most subgroups are much smaller due to lack of seroconversion in the mother and missing samples in the neonates (especially for the first weeks follow-up). For example lines 132, 142, 145... This point makes it more difficult to draw any formal conclusions and authors should discuss this point and underline the limits of their study.

We have updated the discussion to acknowledge the limited sample size of our infant follow up in lines 283-285.

Concerning maternal serology (lines 99-101 and 303), it is not clear whether anti-N and anti-S are detected separately. Moreover, only four patients had serology before vaccination (figure 1); can authors provide clear information on the absence of prior exposure to the virus for the other patients also?

As discussed above. We wanted to use a well-validated serologic assay to evaluate IgG and particularly IgM responses in the cohort. We used the combined RBD/N serology assay that had been well-validated, and we had previously used to detect a maternal-fetal viral transmission by infant IgM response (PMID: 34234001).

Participants were surveyed at enrollment and follow up on prior known COVID-19 infection. No included participants reported a known prior COVID-19 infection during the study time period. In response to these appropriate concerns, we have performed additional experiments including anti-SARS-CoV-2 nucleocapsid antibody screening by ELISA testing in our maternal delivery specimens to evaluate for prior SARS-CoV-2 infection. All included participants were negative by anti-nucleocapsid ELISA testing at delivery. We have updated lines 97-98 and 364-368 to reflect this information.

Lines 218-222: authors rely on the absence of IgM to conclude that there is no transfer of vaccine product. However, IgM are commonly unreliable to diagnose a congenital infection (Kimberlin, JAMA, 2020. Voordouw, Clinical Microbiology Reviews, 2020. Schwartz, Archives of Pathology and Laboratory Medicine, 2020). Please discuss this point.

Early on the pandemic, a number of reports raised concern on performance reliability of IgM serologic testing in the diagnosis of vertical transmission after maternal infection, as discussed in these articles due to unknown sensitivity and specificity of the assays. In the era prior to well validated anti-SARS-CoV-2 IgM assays, this made sense. Consensus and testing are evolving as anti-SARS-CoV-2 IgM testing performance has been better defined. In fact, a large recent review by Allotey et al. in BMJ meta-analysis (PMID: 35296519) review used fetal/infant IgM as one of the defining diagnosis of vertical transmission of SARS-CoV-2. In addition to our work, a number of other studies have found the IgM positivity in infants after maternal infection is reported up to 7% in cohort studies (PMID: 35304596, PMID: 34014840 PMID: 32739398) updated in lines 66-69, 247, and 252-254.

We have since used the presented assay in a number of studies and it has been well-validated as described above. Most importantly, we found a maternal-to-fetal SARS-CoV-2 vertical transmission in which the infant seroconverted to IgM positivity (and rising IgG titers) using the same assay that is used in the presented manuscript (updated in lines 69, 252-254, and 350-353).

Line 223 and 281-282: vaccine products could be safe even if transmitted to the fetus. On the other hand, even if unlikely, vaccine could induce inflammatory response in the mother that could have negative impact on the neonate (Gee, Nat Immunol 2021). Authors don't have enough patients and background that allows driving this conclusion.

We agree with the reviewer on the concern of SARS-CoV-2 infection related inflammation and the potential impact of fetal immune responses as referenced in the important work by Gee et al (PMID: 34616036). The impact of SARS-CoV-2 infection on inflammation and fetal immune responses is also an active area of research in our group. However, inflammation after vaccination was not the focus of this study. Though there are limited studies directly comparing SARS-CoV-2 infection and vaccination during pregnancy, there is no evidence to date that vaccination in pregnancy leads to similar levels of inflammation as infection. Though we are unaware of studies directly examining inflammation after immediately following vaccination in pregnancy, such as plasma cytokines. However, a recent study by Shanes E.D. et al. (PMID: 33975329) evaluated placentas after SARS-CoV-2 vaccination and found no evidence of placental inflammation by histopathologic examination. This is in contrast to SARS-CoV-2 infection during pregnancy which has been associated with high levels of inflammation and abnormal placental histopathology in a number of studies (PMID: 35290925, PMID: 32441303, PMID: 34723226, PMID: 33798476). Additionally, supporting evidence for a lack of biologically relevant inflammation after vaccination may be demonstrated by no increased risk of preterm, still birth, small-gestation-age, or fetal growth restriction in a number of studies of vaccination in pregnancy (PMID: 33882218, PMID: 34990445, PMID: 34496196 PMID: 35323851). In contrast, SARS-CoV-2 infection in pregnancy has been associated with high levels of inflammation, as well as increased risk of stillbirth, preterm birth, and adverse fetal growth restriction in numerous studies. (PMID: 35042863 PMID: 35034863 PMID: 35027756, PMID: 35311909 PMID: 35282784).

Reviewer #3 (Remarks to the Author):

This manuscript reports on 20 people who received an mRNA vaccination during pregnancy, and 19 infants born subsequently. There are four main findings.

1. Following vaccination, IgM is not detected in cord blood. This is in line with previous findings (Mithal, 2021, AJOG; Beharier, 2021, JCI) which should be cited. The authors show, for the first

time, that spike mRNA and protein is not detectable in placenta or cord blood. The authors use these findings to argue that the vaccine does not cross the placenta. Although this is likely to be the case, they will make a stronger argument if they consider the kinetics of vaccine biodistribution and the IgM response. Biodistribution studies (e.g. <https://byrambridle.com/docs/bio-dist-eng.pdf> for BNT162b2) show vaccine clearance within approximately a week, and IgM generally declines below the threshold of detection approximately six weeks after immunization. The authors should focus on those participants who received either their first or their second dose in a timeframe such that vaccine mRNA/protein or IgM would still be detectable if they were present.

We thank the reviewer for the additional references that also show a lack of fetal IgM response after vaccination, and we have updated our discussion with the appropriate citations (Lines 69 and 247).

In our response to reviewer 1, we have added information about timing of vaccination relative to delivery. We have also cited further studies that evaluate the timing and persistence of mRNA vaccine products, and timing of IgM detection as described above.

2. Vaccine-specific IgG, including neutralizing antibodies, are transferred to the infant. This is already well-established - see for example (Gray, AJOG, 2021; Collier, JAMA, 2021; Rottenstreich, CMI, 2021). In particular, it should be noted that a very granular analysis of how timing of vaccination in pregnancy affects umbilical cord blood IgG titers has recently been published (Yang, Obs Gynecol, 2021), which should be cited. The authors claim that functional neutralizing antibodies are differentially transferred to the fetus (Line 177). To assess this claim, we would need to see a comparison between paired transfer ratios for total IgG vs neutralizing antibody, and I cannot see that this has been done. The authors should either do this analysis, or remove the claim.

We thank the reviewer for these important citations that we have updated in lines 259-261, and the important additional analysis suggestion. We have updated Fig 4F with a figure demonstrating the lack of correlation between cord-to-maternal IgG and neutralizing transfer ratios and lines 187-188 and 297. This adds to our conclusion that neutralizing antibodies may be differentially transferred to the fetus.

3. mRNA vaccination leads to a unique SARS-CoV2 Spike protein antibody epitope binding signature, with some of the targeted regions unique to vaccination. However, all the participants in this study were pregnant, whereas none of the participants in the three cited studies on epitope binding following SARS-CoV2 infection were. Although I agree it seems more likely that the key difference is infection versus vaccination, the authors cannot rule out the possibility that the key difference in antibody raised is in fact pregnancy (see for example Atyeo, 2021, Sci Transl Med, who report a subtle difference in the antibody response to the first dose of vaccination during pregnancy). To make this conclusion more firmly, non-pregnant participants should be examined following vaccination. At the very least, this issue should be discussed.

This is an important point. We have expanded our discussion (Lines 215, 220-221, 228, 310-311, 318-327) to include the possibility of difference in functional antibody responses to vaccination in pregnancy, particularly in relation to the study by Atyeo et al, as well as recent work by our collaborators showing potential reduced neutralizing responses to SARS-CoV-2 ancestral virus and variants of concern after mRNA vaccination in pregnancy

(PMID: 35025672).

4. “We believe we are the first to report that infants in the first few months of life continued to have maternal vaccine-derived anti-SARS-CoV-2 antibodies.” Yes, although it’s worth noting that Shook et al have also preprinted this on MedRXiv – this should be cited.

We have added the important work by Shook et al, to our discussion (lines 277-278)

Finally, the authors make a statement to the effect of “timing of vaccination during pregnancy is critical to ensure transplacental transfer of protective antibodies during early infancy” a number of times. This is at odds with the rationale for recommending COVID vaccination during pregnancy, which is to protect mother and baby *during* pregnancy. Suggesting that there is an ideal time in pregnancy to get vaccinated, for the benefit of the infant after it is born, could be counterproductive, in that it may encourage people who are willing to get vaccinated to delay their vaccination, to maximize protection of the newborn. I would suggest changing the emphasis here.

This is a very important point, and we have updated our manuscript to provide further emphasis on the risk of COVID-19 in pregnancy and protection to both the mother and the infant (lines 72-73, 270-273).

REVIEWERS' COMMENTS

Reviewer #1 (Remarks to the Author):

The authors have responded appropriately to all the questions raised in my initial review in this revised submission, and I do not have further concerns. The manuscript will make an important contribution to the field.

Reviewer #3 (Remarks to the Author):

I am satisfied that the authors have responded to my major points and will now recommend the manuscript for publication.

Reviewer comments:

Reviewer #1 (Remarks to the Author):

The authors have responded appropriately to all the questions raised in my initial review in this revised submission, and I do not have further concerns. The manuscript will make an important contribution to the field.

Reviewer #3 (Remarks to the Author):

I am satisfied that the authors have responded to my major points and will now recommend the manuscript for publication.

Response to reviewers:

We thank the reviewers for their critical insights and valuable feedback. We appreciate the positive comments.